# Transoral Robot-Assisted Surgery in Supraglottic and Oropharyngeal Squamous Cell Carcinoma: Laser Versus Monopolar Electrocautery

**DOI:** 10.3390/jcm8122166

**Published:** 2019-12-07

**Authors:** Marco Benazzo, Pietro Canzi, Simone Mauramati, Fabio Sovardi, Antonio Occhini, Eugenia Maiorano, Giuseppe Trisolini, Patrizia Morbini

**Affiliations:** 1Department of Otorhinolaryngology, University of Pavia, IRCCS Policlinico San Matteo Foundation, 27100 Pavia, Italy; marco.benazzo@unipv.it (M.B.); simone.mauramati@gmail.com (S.M.); sovardi.fabio@gmail.com (F.S.); antonio.occhini@alice.it (A.O.); eugenia_maiorano@libero.it (E.M.); giuseppe.trisolini01@universitadipavia.it (G.T.); 2Unit of Pathology, Department of Molecular Medicine, University of Pavia, IRCCS Policlinico San Matteo Foundation, 27100 Pavia, Italy; patrizia.morbini@unipv.it

**Keywords:** transoral robotic surgery, thulium laser, CO_2_ laser, cancer

## Abstract

Background: Monopolar electrocautery (EC) is the surgical cutting and haemostatic tool most commonly used for transoral robotic surgery (TORS). The aim of this study was to retrospectively compare EC efficacy in the treatment of patients affected by T1 or T2 oropharyngeal and supraglottic squamous cell carcinomas with the more recently introduced laser fibres. Methods: We considered all TORS patients admitted to our department from January 2010 to June 2019. The outcomes of patients treated with Thulium: yttrium aluminium garnet (YAG) laser (TY-TORS), CO_2_ laser (CO_2_-TORS) and EC (EC-TORS) were analysed in order to assess surgical performances, functional outcomes and postoperative complications. Results: Twenty patients satisfied the enrolling criteria, of which nine underwent laser-TORS, and the remaining 11 underwent EC-TORS. In all candidates, TORS procedures were completed without the need for microscopic/open conversion. Close or positive margins were significantly more frequent in EC-TORS (*p* = 0.028). A considerable difference was found in overall functional parameters: times of nasogastric tube and tracheostomy removal and time of hospital discharge were significantly shorter in laser-TORS (*p* = 0.04, *p* = 0.05, *p* = 0.04, respectively). Conclusions: Laser-TORS showed better results in comparison with EC-TORS in term of tumour resection margins and patient functional outcomes. Our findings can be justified with the greater tissue thermal damage caused by EC-TORS, despite prospective randomized trials and increased patient numbers being needed to confirm these preliminary conclusions.

## 1. Introduction

Since its first introduction in head and neck surgery [1], the transoral robotic approach has represented an effective and validated option in the management of T1 or T2 oropharyngeal and supraglottic tumours [2,3]. The comparison of the relative outcomes of the different therapeutic strategies available to treat these cancers (robotic surgery, transoral laser microsurgery, chemo-radiotherapy) is important to assess the best disease management and the optimal functional organ preservation for each case. Wide and 3D operative visualization, multidirectional and angled surgical motion, and intuitive and fast learning use are only some of the many advantages of transoral robotic surgery (TORS). Monopolar electrocautery is the most commonly used surgical cutting and haemostatic tool for TORS [1]. However, as a consequence of the limits of this instrument, new tools have been implemented in recent years [4]. In fact, the relatively high thermal damage and tissue necrosis related to the monopolar electrocautery (EC) may negatively influence the oncologic and functional results in terms of tumour resection, quality of life and postoperative complications [4,5]. The recent development of flexible laser fibres has at last extended their use to TORS, with the aim of overcoming most EC limits [6]. Diode thulium: yttrium aluminium garnet (YAG) laser (TY) has been the first to be approved to be coupled with robotic technology [4]. We previously explored the feasibility of flexible thulium laser coupling with a novel robotic introducer for the treatment upper aerodigestive cancers [7]. More recently, flexible carbon dioxide (CO_2_) laser fibres have been developed to be suited for TORS, in order to reduce the severity of thermal damage. In current literature, few clinical trials have compared the outcomes of laser-TORS (CO_2_-TORS or TY-TORS) with EC-TORS [8,9]. Up to now, only one study reported data concerning surgical margins status [8]. The aim of the present study was to retrospectively assess laser benefits and outcomes in comparison with EC in patients affected by T1 or T2 oropharyngeal and supraglottic squamous cell carcinoma submitted to TORS.

## 2. Materials and Methods

We retrospectively analysed all patients who underwent TORS for treatment of pharyngolaryngeal tumours at the Otorhinolaryngology Department of Policlinico S. Matteo and University of Pavia (Italy) from January 2010 to June 2019, following institutional review board approval. Inclusion criteria are summarised as follows:All patients consecutively scheduled for TORS with laser (TY- or CO_2_- laser) or EC, performed by the same surgeon team (M.B. and A.O.).cT1/T2 oropharyngeal and supraglottic squamous cell carcinomas, cN0/2 (according to the American Joint Committee on Cancer (AJCC) Cancer Staging Manual 8th edition) [10].cM0 at oncological staging including panendoscopic biopsies of the upper aerodigestive tract, head and neck/chest Computed Tomography (CT) scan, and Magnetic Resonance Imaging (MRI) as needed.

All patients were preoperatively counselled about the conventional alternatives and signed informed consent was obtained. Surgical robotic set-up was predisposed as previously reported [7], and included:-An Intuitive da Vinci S System (Intuitive Surgical, Inc., Sunnyvale, CA, USA);-An Intuitive Surgical^®^ Endo Wrist Introducer, 5Fr, to hold and position the thulium surgical laser fibres (Revolix Jr, LISA Laser, Katlenburg-Lindau, Germany). The power setting of a 2 µm continuous-wave TY ranged between 5.0 and 8.5 W during respectively margin incision and lesion removal;-Flexible CO_2_ Laser Fibre (OmniGuide Surgical, Lexington, MA, USA) inserted in a flexible metal carrier (grasped with Maryland atraumatic forceps). The CO_2_ laser power was set to 14 W for dissection and 7 W for coagulation;-A 5-mm monopolar EC (Intuitive Surgical, Erbotom ICC 350 ERBE Elektromedizin GmbH, Tübingen, Germany);-A 5-mm Maryland forceps (EndoWrist; Intuitive Surgical, Inc);-A Feyh-Kastenbauer retractor (Gyrus Medical Inc., Maple Grove, MN, USA).

The cutting tool (TY-laser, CO_2_-laser or EC) was handled in the left or right robotic arm relative to patient anatomy, independently of the surgeon handedness.

Recorded outcomes were analysed in order to assess three categories of results: surgical performances (surgical robotic time (SRT), estimated blood loss (EBL), status of tumour resection margins at histopathological evaluation, need for microscopic/open conversion), functional outcomes (oral diet recovery, time to tracheostomy removal, time to discharge), and postoperative complications. Adjuvant therapy included intensity modulated radiotherapy with or without chemotherapy. Indications for these treatments were the presence of neck disease with multiple positive lymph nodes, extracapsular extension, and atypical metastatic patterns. Indications related to the primary tumour included histopathological evidence of lymphovascular invasion, positive resection margins, and perineural invasion. The decision for adjuvant therapy was based on multidisciplinary tumour board discussion of the case.

Medcalc Version 19.0.5 (MedCalc Software, Ostend, Belgium) was used for statistical analysis. Continuous variables were compared using a Kruskal–Wallis test. Categorical data were presented as frequencies and compared, using Fisher’s exact test to obtain the *p* value.

## 3. Results

Twenty patients (10 men and 10 women) aged 38–84 years (mean = 61.7 years) underwent TORS for supraglottic and oropharyngeal squamous cell carcinoma (Table 1). Nine patients underwent laser-TORS, two for supraglottic and seven for oropharyngeal tumours. Only one patient, affected by an oropharyngeal squamous cell carcinoma, underwent TORS surgery with CO_2_ laser, while the other eight patients were treated with thulium:YAG laser TORS. The remaining 11 patients underwent EC-TORS, two for supraglottic and nine for oropharyngeal tumours (Table 2 and Table 3).

The two patient groups had comparable TNM tumour stage distribution but for supraglottic cancers. The TY-TORS group included only T1 cancers. Each surgical procedure was performed under general endotracheal anesthesia. All approaches were completed without the need for conversion to a microscopic/open procedure. Unilateral or bilateral selective neck dissection was performed using an open approach, in all cases, at the same time as TORS: eight patients were subjected to unilateral selective neck dissection and 10 to bilateral neck dissection. While nine patients were only treated with surgery, 11 patients needed adjuvant treatment (radiotherapy in five cases, chemotherapy in two, chemo-radiotherapy in four). In the laser-TORS group, no intraoperative adverse event occurred in patients treated for oropharyngeal squamous cell carcinoma; good bleeding control was always achieved. Only one postoperative complication was observed in this group, consisting of postoperative bleeding in a patient treated for squamous cell carcinoma (SCC) of the tongue base. Bleeding was stopped with electrocautery. Complications related to EC-TORS included three postoperative oropharyngeal hemorrhages and one intraoperative pharyngotomy (treated harvesting a sternocleidomastoid pedicled muscle flap).

In patients treated with laser-TORS for supraglottic SCC no operative/postoperative complications were reported; in those treated with EC-TORS, one postoperative cardiac arrest was observed, which resulted in a longer postoperative recovery time.

In the laser-TORS group, the mean follow-up for patients treated for oropharyngeal SCC was 77.4 months (±31.5 months). No patient was lost to follow-up. In the EC-TORS group the mean follow-up was 57.6 months (±24.0 months); one patient died, and one was lost to follow up. In patients treated for supraglottic SCC, the mean follow-ups were 105 months (±9.9 months) for TY-TORS and 57 months (±12.7 months) for EC-TORS. No patient was lost to follow-up. All patients were alive without evidence of disease at the time of last follow-up. Comparative outcomes focusing on main surgical and clinical variables are summarized in Table 4 and Table 5.

The comparison of outcome measures was possible between global laser-TORS and EC-TORS populations; comparison was also possible in the subgroups of patients affected by oropharyngeal SCC. The comparison between the two TORS modalities was not possible for patients affected by supraglottic carcinoma due to the insufficient number of patients.

The average length of robotic excision and the estimated blood loss (EBL) were similar in EC and laser groups. The average hospital stay was significantly shorter for laser-TORS patients than for EC-TORS (9.8 vs. 13.8 days, *p* = 0.05).

The mean times to nasogastric tube and tracheotomy removal were 5 and 7.2 days in laser group, respectively, and 8.7 and 9.8 days in EC group, respectively (*p* = 0.04).

Close or positive margins were observed more frequently in the EC group than in laser group (*p* = 0.028): in the EC-TORS group, four patients had positive resection margins (three oropharyngeal SCC and one supraglottic SCC), and two patients had close (i.e., <5 mm) resection margins; in laser-TORS only one patient had positive margin. No significant difference in complication rates have been observed.

When we compare oropharyngeal resection to supraglottic laryngectomy, the surgical robotic time resulted significantly longer for the second procedure (*p* = 0.002). Moreover, patients treated with supraglottic laryngectomy had a significantly longer hospital stay (*p* = 0.013). No differences were recorded in EBL and time to nasogastric tube and tracheostomy removal.

## 4. Discussion

Laser technology has represented an important step forward for head and neck surgery. In spite of the considerable advantages fulfilled by this technology, linear beam cutting has initially represented the main limitation to a more widespread use of this instrument. The introduction of flexible optical fibres capable of conveying this type of energy has allowed to expand laser applications and to integrate them with robot-assisted surgery.

Robotic surgery has been widely adopted across several specialties thanks to its numerous advantages, including shorter operating times, shorter hospital stays, and fewer intra- and postoperative complications as compared with traditional surgery. One of the main fields of application of robotic surgery in head and neck pathology is TORS [1,11]. The great oncological and functional results obtained by TORS in the treatment of early pharyngolaryngeal cancers have strongly encouraged its spread, providing numerous confirmations of feasibility [7,12,13], but also highlighting its limitations and the need for new tools to achieve even better outcomes [14].

Different cutting and haemostatic tools have been developed in order to limit tissue injury and induce a favourable wound healing process. Hoffman et al. [4] evaluated and compared the performances of four different instruments (Laser CO_2_, Tm:YAG laser, monopolar electrocautery, radiofrequency needle) on a porcine model, focusing on the width of incision and coagulation zones, bleeding, tissue sticking, user friendliness, speed and costs. However, the comparison of functional and oncological results obtained by different tools in surgical practice is a fundamental step for a conscious preoperative evaluation of the cost–benefit ratio. Current literature provides a limited number of studies dealing with this topic. In particular, Karaman et al. [9], performed a systematic and quantitative comparison between CO_2_-TORS and EC-TORS in a population of 20 patients who underwent tongue base resection for sleep apnoea syndrome. In their study, laser-TORS resulted in less intraoperative bleeding, shorter robot operating time, shorter lengths of hospitalization, shorter feeding rehabilitation and less postoperative pain, when compared with EC-TORS. Previously, Van Abel et al. [8] reported a comparison of TY-TORS (15 patients) with EC-TORS (30 patients); the use of thulium laser resulted in less postoperative pain, which may be due to decreased collateral thermal damage, and finer cutting [8].

In our study, we compared EC-TORS with laser-TORS focusing not only on functional but also on oncological results in a population of 20 patients affected by pharyngolayngeal cancers. Despite the limits intrinsic to a retrospective type of analysis, our findings provided some interesting data. First of all, both approaches allowed successful completion of the surgical procedure in all patients, without requirement for microscopic or open conversion. SRT and EBL data were similar to those reported in literature and did not differ among the two groups [8,9,12], as well as rates of postoperative bleeding and frequency of airway complications. Positive margins were more frequent in EC-TORS, even though they did not seem to impact on survival or recurrence rate. However, considerable differences in overall functional parameters were observed, including time of discharge, and times of nasogastric tube and tracheotomy removal, which were significantly shorter in laser-TORS. This result confirms previous observations in a non-oncological setting [9].

The reported differences between laser and EC-TORS can be related to the different electrocautery properties of the two cutting modalities: the greater thermal energy delivered by EC-TORS and the resulting tissue necrosis can justify the worse functional parameters observed in comparison with laser-TORS. The higher rate of positive margins on histology may be due to a more extensive loss of tissue in consequence of energy transfer, which reduces the extension of “safe tissue” that can be observed at the periphery of the neoplastic lesion on the resected specimen.

In order to confirm these initial findings, further prospective randomized trials and a large number of patients are needed, in particular to analyse the subgroup of supraglottic cancers, whose number was too limited for statistical evaluation.

## 5. Conclusions

Despite the limited number of cases available for the study, laser-TORS showed better results in comparison with EC-TORS in term of tumour resection margins and functional outcomes. Less severe thermal injury and more precise cutting properties of flexible laser fibres may offer significant advantages in transoral treatment of early stage oropharyngeal and supraglottic cancers.

## Figures and Tables

**Table 1 jcm-08-02166-t001:** Demographic and clinical data of the patients.

Patient No.	Robotic Resection Tool	Years	Sex	Tumour Subsites	Histology	cTNM (AJCC)	Neck Dissection	Preceding Months	Subsites	Histology	TNM (AJCC)	Therapy	IID cm
1	TY	62	F	Epiglottis	SCC G2	c T1N0M0							4.3
2	TY	54	F	Epiglottis	SCC G1	c T1N2cM0	L,R, II-III-IV						4.0
3	TY	48	M	L, lateral oropharyngeal wall	SCC G3	c T1N0M0	L, I-II-III-IV	24	L tonsil and L tongue base	SCC G3	pT2N0M0	RT-CH	4.5
4	TY	55	M	R, tongue base	SCC G2	c T1N0M0	L,R, I-II-III-IV	24	L, hypopharynx	SCC G3	pT2N1M0	RT-CH	4.0
5	TY	68	F	R, tongue base	SCC G2	c T1N1M0	R,L, I-II-III-IV						4.5
6	TY	56	F	L, tongue base	SCC G2	c T1N0M0	R, I-II-III-IV	24	L neck	SCC G2	pTxN1M0	L ND	3.5
7	TY	65	F	L, tongue base	SCC G2	c T2N0M0		24	L, submandibular gland	Adenoid-cystic ca	pT1N0M0	L, Scialoadenectomy, ND	4.0
8	TY	58	M	L, tongue base	SCC G2	c T2N1M0	L, II-III-IV						5.0
9	EC	48	M	R, lateral oropharyngeal wall	SCC G2	c T1N0M0		24	Larynx	SCC G2	pT2N0M0	Supraglottic laryngectomy,	3.5
												L,R ND	
10	EC	52	F	L, tonsil	SCC G2	c T1N1M0	L, I-II-III-IV						5.5
11	EC	84	M	L, tonsil	SCC G2	c T2N2bM0	L, I-II-III-IV						4.0
12	EC	56	M	R, tongue base	SCC G2	c T2N2bM0	R, I-II-III-IV						4.5
13	EC	61	F	R, tonsil	SCC G2	c T2N1M0	R,L,I-II-III-IV						3.5
14	EC	72	M	R, tonsil	SCC G2	cT2N1M0	R,II-III-IV						5
15	EC	77	M	R, tonsil	SCC G2	c T1N1M0	R, II-III-IV						3.5
16	EC	55	F	Epiglottis	SCC G2	c T2N2aM0	R,L II-III-IV						4.3
17	EC	79	M	Epiglottis	SCC G2	c T1N0M0							4.0
18	EC	72	F	R, tongue base	SCC G2	CT1N0M0					pT1N0M0		4.3
19	EC	52	F	L, tonsil	SCC G2	CT1N0M0					pT1N0M0	RT	4.5
20	CO_2_	59	M	R, tonsil	SCC	cT2N1M0	R,II-III-IV				pT3N1M0		4.3

M = male, F = female, L = left, R = right, IID = interincisor distance, ND = neck dissection, TY = Thulium: yttrium aluminium garnet (YAG) laser, EC = monopolar electrocautery, SCC = squamous cell carcinoma, RT = radiotherapy, CH = chemotherapy, AJCC = American Joint Committee on Cancer.

**Table 2 jcm-08-02166-t002:** Patients’ operative and clinical data (oropharynx).

Robotic Resection Tool	Patient No.	SRT min	EBL mL	Tumour Subsites	Oral Diet, Days	Time to Tracheostomy Removal, Days	Time to Discharge Days	Complications	Margins	Postoperative Treatment	pTNM	Follow-Up, Months
TY	3	55	10	L, lateral oropharyngeal wall	4	7	7		negative	NO	p T1N0M0	96
TY	4	40	30	R, tongue base	5	7	8	Day 22: bleeding stopped with electrocautery control	negative	NO	p T1N0M0	85
TY	5	50	10	R, tongue base	4	5	7		negative	RT	P T1N1M0	82
TY	6	35	25	L, tongue base	6	11	13		negative	NO	p T1N0M0	78
TY	7	40	30	L, tongue base	6	7	9		negative	NO	p T2N0M0	76
TY	8	50	40	L, tongue base	3	6	7		negative	CH-RT	p T2N1M0	69
CO_2_	20	40	20	R, tonsil	6	7	8		positive	CH-RT	pT3N1M0	1
EC	9	50	45	R, lateral oropharyngeal wall	8	12	13		close	CH	P T1N0M0	98
EC	10	40	35	L, tonsil	6	7	10		negative	RT-CH	p T1N1M0	95
EC	11	50	25	L, tonsil	5	4	19		positive	RT	p T2N2bM0	52
EC	12	55	35	R, tongue base	9	13	13		close	CH-RT	p T2N2bM0	43
EC	13	40	30	R, tonsil	9	9	11	Intraoperative pharyngotomy Day 12: bleeding stopping with electrocautery control	negative	CH	p T2N1M0	70
EC	14	45	10	R, tonsil	2	9	9	Day 11: bleeding stopped with electrocautery control	positive	RT	pT2N2aM0	57
EC	15	40	45	R, tonsil	5	7	8		negative	RT	p T1N1M0	54
EC	18	45	25	R, tongue base	15	16	20		positive	open surgery	pT1N0MO	18
EC	19	35	35	L, tonsil	9	9	11		negative		pT1N0MO	33

L = left, R = right, SRT = surgical robotic time, EBL = estimated blood loss, RT = radiotherapy, CH = chemotherapy.

**Table 3 jcm-08-02166-t003:** Patients’ operative and clinical data (larynx).

Robotic Resection Tool	Patient No.	SRT min	EBL mL	Tumour Subsites	Oral Diet, Days	Time to Tracheostomy Removal, Days	Time to discharge, days	Complications	Margins	Postoperative Treatment	pTNM	Follow-Up, Months
TY	1	135	35	Epiglottis	6	8	15		negative	NO	pT1N0M0	112
TY	2	110	20	Epiglottis	5	7	14		negative	NO	PT1N0M0	98
EC	16	130	40	Epiglottis	10	12	14		positive	CH-RT	pT2N1M0	66
EC	17	120	30	Epiglottis	18	20	24	Day 7: cardiac arrest	negative	NO	pT1N0M0	48

SRT = surgical robotic time, EBL = estimated blood loss, RT = radiotherapy, CH = chemotherapy, TY = Thulium: yttrium aluminium garnet (YAG) laser, EC = monopolar electrocautery.

**Table 4 jcm-08-02166-t004:** Laser transoral robotic surgery (laser-TORS) versus monopolar electrocautery (EC)-TORS (oropharynx).

Robotic Resection Tool	SRT, min (Mean)	EBL, mL (Mean)	Oral Diet, Days (Mean)	Time to Tracheostomy Removal, Days (Mean)	Time to Discharge, Days (Mean)	Postoperative Complications		Tumour Margins		
						Yes	No	Positive	Negative	Close
Laser-TORS (*n* = 7)	44.3 ± 7.3	23.6 ± 11.1	4.9 ± 1.2	7.1 ± 1.9	8.4 ± 2.1	1	5	1	6	0
EC-TORS (*n* = 9)	44.4 ± 6.3	31.7 ± 10.9	7.5 ± 3.7	9.5 ± 3.6	12.7 ± 4.2	2	5	3	4	2

SRT = surgical robotic time, EBL = estimated blood loss, laser-TORS = Laser transoral robotic surgery, EC-TORS = monopolar electrocautery transoral robotic surgery

**Table 5 jcm-08-02166-t005:** TY-TORS versus EC-TORS (larynx).

Robotic Resection Tool	SRT, min (Mean)	EBL, mL (Mean)	Oral Diet, Days (Mean)	Time to Tracheostomy Removal, Days (Mean)	Time to Discharge, Days (Mean)	Postoperative Complications		Tumour Margins		
						Yes	No	Positive	Negative	Close
TY-TORS (*n* = 2)	122	27.5	5.5	7.5	14.5	0	2	0	2	0
EC-TORS (*n* = 2)	125	35	14	16	19	1	1	1	1	0

SRT = surgical robotic time, EBL = estimated blood loss, TY-TORS = Thulium:(YAG laser transoral robotic surgery, EC-TORS = monopolar electrocautery transoral robotic surgery

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
