# Peer review of "Transoral Robot-Assisted Surgery in Supraglottic and Oropharyngeal Squamous Cell Carcinoma: Laser Versus Monopolar Electrocautery"

_jcm, 2019, doi:10.3390/jcm8122166_

Round 1

Reviewer 1 Report

This paper is a comparative study of laser vs. monopolar electrocautery in TORS. Laser assisted TORS showed better results than electrocautery assisted TORS in terms of tumor resection margins and patient's quality of life. The results and methods of research is profitable to surgeons, and are mandatory for publication. However, minor revision should be made.

In abstract, "Recently laser fibres became suitable too, the aim of this study is to 16 compare the efficacy of these instruments in patients" should be "~ too. The aim".

Abbreviations in the abstract also needs explanations, such as "TY-TORS" and "CO2-TORS".

It would be better to clarify methods for neck dissection. Was it performed by an open procedure or by a robotic neck dissection. What incision did you use for a neck dissection?

In table 1, there is two columns with the same subtitle "histology". It would be better to put an additional explanation for this.

Author Response

We thank the Reviewer for the time devoted to our research and for the positive evaluation of the content. In the following we address all the issues raised, highlighting the modification to the manuscript.

In abstract, "Recently laser fibres became suitable too, the aim of this study is to 16 compare the efficacy of these instruments in patients" should be "~ too. The aim".

Along with the revision of the English language, we also edited this sentence. (Abstract, page 1, lines 24-27).

Abbreviations in the abstract also needs explanations, such as "TY-TORS" and "CO2-TORS".

We explain the abbreviations "TY-TORS" and "CO2-TORS", as You suggested. (Abstract, page 1, lines 28-29).

It would be better to clarify methods for neck dissection. Was it performed by an open procedure or by a robotic neck dissection. What incision did you use for a neck dissection?

As You suggested, the type of surgical approach for neck dissection has been specified (Results, page 3 line 128-131)

In table 1, there is two columns with the same subtitle "histology". It would be better to put an additional explanation for this.

We deleted one of the two columns because it represented a duplicate and was inserted by mistake. Thank You for the correction. (Table 1, pages 7-8)

Reviewer 2 Report

Dear authors, 

Firstly, an extensive editing of the English language is required and it should be performed by a native English speaker. I recommend correcting the syntax and using shorter sentences. 

Furthermore, I recommend the following changes throughout the manuscript:

TITLE

Please remove the abbreviation (TORS)

I would recommend the title as following:

Transoral robot-assisted surgery in supraglottic and oropharyngeal squamous cell carcinoma: laser versus monopolar electrocautery

ABSTRACT

Use abbreviations when mentioning the word for the first time (TORS, line 16).

Add "," after recently in the line 15.

Add "." after "too" in line 15 and then beginn the new sentence.

You described analyzing patients' quality of life. However, the outcomes that were analyzed in that regard (oral diet recovery, time to tracheostomy etc.) are rather short-term outcomes. Quality of life would mean analyzing questionaries, for example. Please rephrase (in Abstract and throughout the manuscript).

INTRODUCTION

Rephrase the last sentences. It should be more clear why this study was conducted in regards to the current literature.

DISSCUSION

I would recommend a rewrite of this section, especially from line 172 to line 190. When citing studies with similar design, please provide more details on their results and compare them to yours. 

Moreover, more detailed discussing of your own results should be provided.

CONCLUSION

Again, rephrasing the part about "quality of life" is needed.

Author Response

We thank the Reviewer for the time devoted to our research and for the positive evaluation of the content. In the following we address all the issues raised, highlighting the modification to the manuscript.

Firstly, an extensive editing of the English language is required and it should be performed by a native English speaker. I recommend correcting the syntax and using shorter sentences. 

Thank you for highlighting this issue. As requested, we proceeded to an extensive editing of the English language.

     2. Title: please remove the abbreviation (TORS)

We sincerely appreciated your suggestion. The title has been changed to: “Transoral robot-assisted surgery in supraglottic and oropharyngeal squamous cell carcinoma: laser versus monopolar electrocautery” (Title, lines 2-7)

Use abbreviations when mentioning the word for the first time (TORS, line 16).

We explained all the abbreviations as requested (Abstract, page 1, line 22).

You described analyzing patients' quality of life. However, the outcomes that were analyzed in that regard (oral diet recovery, time to tracheostomy etc.) are rather short-term outcomes. Quality of life would mean analyzing questionaries, for example. Please rephrase (in Abstract and throughout the manuscript).

Actually, the data we collected evaluate the functional outcomes during the postoperative course. Therefore, we have replaced "quality of life" with "functional outcomes" throughout the manuscript.

Introduction: Rephrase the last sentences. It should be more clear why this study was conducted in regards to the current literature.

We emphasized that few clinical trials have compared the outcomes of laser-TORS with EC-TORS.  In particular, only one study reported data concerning surgical margins status in cancer resection while the other one takes into account tongue base resections in OSAHS patients.  (Introduction, page 2, lines 65-70)

Discussion: I would recommend a rewrite of this section, especially from line 172 to line 190. When citing studies with similar design, please provide more details on their results and compare them to yours.

        Moreover, more detailed discussing of your own results should be provided.

The discussion has been revised providing more details on the cited studies results.

Conclusion: Again, rephrasing the part about "quality of life" is needed.

The conclusions have been rephrased.